# Simultaneous Re-Planning and Plan Execution for Online Job Arrival

**Ivan Gavran[1], Maximilian Fickert[2], Ivan Fedotov[1], Jörg Hoffmann[2], Rupak Majumdar[1]**

[1]Max Planck Institute for Software Systems, Kaiserslautern, Germany

[2]Saarland University, Saarland Informatics Campus, Saarbrücken, Germany

{gavran, ivanan, rupak}@mpi-sws.org, {fickert, hoffmann}@cs.uni-saarland.de

## Abstract

In many AI planning applications, an agent receives new jobs (additional non-conflicting goals) while plan execution is still ongoing. Vanilla solutions are to (a) finish execution before tackling the new job, or to (b) interrupt execution and re-plan immediately. Option (a) misses opportunities to smoothly integrate the new job into the plan, while (b) leaves the agent idle during re-planning. We introduce *simultaneous re-planning and execution* (SRE), a planning algorithm that avoids both disadvantages. SRE re-plans for both the old and new jobs while the current plan is still being executed. The key difficulty is that, then, the initial state for the revised plan–the state in which plan execution is at the end of re-planning–depends on the time taken for re-planning. We address this through a variant of A$^*$ that starts with several speculative initial states, and incorporates time-aware search information to differentiate between these. On a collection of extended planning competition benchmarks, our algorithm consistently outperforms both (a) and (b).

## Introduction

In AI planning (Ghallab, Nau, and Traverso 2004), the task is to find a schedule of actions leading from the initial state of an agent to a goal state. Planning tools are given a description of states, actions, and goal as input, and should automatically produce a plan. While planning is often seen as a discipline of "thinking before acting," many problems require a constant interplay between thinking and acting (Myers 1999; Ghallab, Nau, and Traverso 2016): e.g., Mars rovers (Estlin et al. 2000; Knight et al. 2001), high-speed manufacturing (Ruml, Do, and Fromherz 2005), or modular printer controllers (Ruml et al. 2011). Here, we address *continual online planning* (Lemons et al. 2010; Burns et al. 2012), where an agent has to tackle a changing set of goals. We consider a special case we call *continual online job arrival*, where the new goal is akin to an additional *job*, that does not conflict with the previous goal.

Similar problems have been considered in different circumstances and under different assumptions. Agents with knowledge about the goal arrival distribution can anticipate future goals and plan accordingly (Burns et al. 2012). If the new goals are known to be similar to the old ones, plan repair can be used (Fox et al. 2006). Here we make neither of these assumptions, tackling arbitrary new goals/jobs arriving online.

There are two vanilla solutions for an online plan execution and re-planning loop in our context: (a) keep executing the current plan to its end before starting the execution of the (re-planned) new plan incorporating the new job; or (b) interrupt the execution of the current plan and wait for the re-planning process to finish. Both strategies have pros and cons. Option (a) allows (some of) the re-planning to be done in parallel to the execution. However, while moving towards the old goal, the agent might be moving away from the new goal, thus missing the opportunity to smoothly incorporate the new job into the current plan. Option (b) takes this opportunity, but leaves the agent idle for the entire re-planning process, which is wasteful when re-planning takes a long time. In this paper we direct our attention towards scenarios where the planning time can not be assumed to be negligible with respect to execution time.

We propose a *simultaneous re-planning and executing* algorithm (that we will refer to as *SRE*) that plans for both the old and the new task while executing the current plan, thus combining the advantages of both previously described strategies. This raises a new challenge: the agent changes its state while re-planning, so the initial state for the revised plan depends on the time taken by the re-planning process. That process must thus be aware of, and reason about, its own duration in order to determine the initial state for the revised plan.

Planners able to reason about their own planning time are called *time-aware* (e.g., (Burns, Ruml, and Do 2013; Cashmore et al. 2018)). Unlike classical planners, which optimize plan cost (e.g. the plan duration), time-aware planners use the (estimated) time needed for planning as a part of their cost function. To illustrate: a user cares about the time needed to get a cup of coffee from a robot; what fraction of that time was spent planning, and what fraction was spent executing the plan, is irrelevant to the user. Time-aware planners do not solve our challenge here as they still assume a fixed initial state. However, as we shall see, the techniques used by time-aware planners can help address that challenge in our context.

Our SRE algorithm is a variant of A$^*$ (Hart, Nilsson, and Raphael 1968) that takes advantage of time-aware techniques to adapt to the setting with asynchronous task arrivals. There are two high-level differences between SRE and A$^*$.

First, while A\* starts its search from a single initial state, SRE starts with a number of potential initial states. These states represent a speculation on the state in which the agent might be once the planning is done. Second, A\*'s ordering function is extended with an additional heuristic function, estimation of when the planning process will finish, informing the search about which of the possible initial states, and search paths below these, is promising to explore. We prove that, under suitable conditions on the heuristic functions used by SRE, the first solution it finds is better than any other it might find by continuing the search. This property and its proof correspond to a similar property of the time-aware search algorithm Bugsy (Burns, Ruml, and Do 2013).

Overall, our contributions are

- a time-aware search algorithm for online AI planning with asynchronous task arrival;

- an implementation of the algorithm in Fast Downward (Helmert 2006), leveraging time-aware planning techniques (Dionne, Thayer, and Ruml 2011) as well as heuristic search planning techniques (Hoffmann and Nebel 2001);

- an empirical comparison of the algorithm to two baselines on a collection of benchmarks from the international planning competition (IPC), which we extended for our online setting; SRE consistently outperforms both vanilla solutions.

## Problem definition

We formulate our setting as a variant of classical planning with online job arrivals during an ongoing execution. The focus is on the moment when a new task arrives while the agent is already executing its *current plan*. This generalizes straighforwardly to a series of arriving jobs, assuming that they arrive once the planning phase is done.

### Background

We consider the *finite-domain representation (FDR)* (Bäckström and Nebel 1995; Helmert 2009) for classical planning tasks:

**Definition 1.** A **planning task** is a tuple $(V, A, c, s_0, s_*)$:

- $V$ is a finite set of *state variables*, each with a finite domain of possible values,

- $A$ is a finite set of *actions*. Each action $a$ is a pair $(pre_a, eff_a)$ of partial variable assignments called *preconditions* and *effects*,

- $c\colon A \to \mathbb{R}$ is a function assigning a cost to every action,

- $s_0$ is a complete variable assignment called *initial state*,

- $s_*$ is a partial variable assignment called *goal*.

We denote the set of all complete variable assignments, or *states*, by $\mathcal{S}$. A partial assignment $p$ is said to be *compliant* with a state $s \in S$ (denoted by $p \subseteq s$) if there is no variable in the domain of $p$ to which $p$ and $s$ assign different values. An action $a \in A$ can only be applied to a state $s \in S$ if $pre_a \subseteq s$. The outcome of that application is state $s[\![a]\!]$, that

is the same as $s$, except that the variables in the domain of partial assignment $eff_a$ are changed accordingly.

A solution (*plan*) to a planning task is a sequence of actions $\overline{a_1, a_2, \ldots, a_n}$ with the overall cost $C(\overline{a_1, a_2, \ldots, a_n}) = \sum_{i=1}^{n} c(a_i)$, leading from $s_0$ to a state compliant with $s_*$.

In what follows, the costs of actions (function $c$) will be interpreted as the duration to execute them. We do not, however, consider concurrent plans as in temporal planning (Fox and Long 2003), limiting our focus to sequential plans with action durations instead. Exploring concurrent temporal planning remains an important topic for future work.

We are considering tasks where the set of goals is not fixed, and new goals may appear online. This has been called *continual online planning (COP)* (Lemons et al. 2010; Burns et al. 2012). COP tasks have been defined as Markov Decision Processes where additional goals may arrive at each time step, and world states are extended with the current goal set. We adapt this notion to COP tasks as classical planning tasks that are extended with a second goal condition, assumed to arrive during the execution of the plan for the original goal.

**Definition 2.** A **continual online planning (COP) task** is a tuple $(V, A, c, s_*^{\text{OLD}}, s_*^{\text{NEW}}, s_0, \pi_{s_0, s_*^{\text{OLD}}})$ where

- $V$ is a finite set of *state variables*, each with a finite domain of possible values.

- $A$ is a finite set of actions. Each action $a$ is a pair $(pre_a, eff_a)$ of partial variable assignments,

- $c\colon A \to \mathbb{R}$ is a function that assigns a cost to every action $a \in A$. We interpret the cost $c(a)$ as the duration needed to execute action $a$,

- $s_*^{\text{OLD}}$ is a partial variable assignment called *old goal*,

- $s_*^{\text{NEW}}$ is a partial variable assignment called *new goal*,

- $s_0$ is the state denoting the agent's position at the time when $s_*^{\text{NEW}}$, the new goal, appeared,

- $\pi_{s_0, s_*^{\text{OLD}}} = \overline{a_1 a_2 \ldots a_n}$ is a sequence of actions, taking the agent from the state $s_0$ to a state compliant with the old goal $s_*^{\text{OLD}}$ (the agent's *current plan*).

A solution to a COP task is a plan $\pi$ consisting of two parts: a prefix of $\pi_{s_0, s_*^{\text{OLD}}}$ and the newly planned extension. There must exist $1 \le j \le n$ such that $\pi = \overline{a_1 a_2 \ldots a_j b_1 \ldots b_m}$ (where the extension, denoted by actions $b_1$ to $b_m$, can be empty). The state (partial assignment) to which the plan $\pi$ takes the agent must be compliant with both $s_*^{\text{OLD}}$ and $s_*^{\text{NEW}}$. A solution is said to be optimal if it minimizes the overall planning and execution time, i.e., the time from the arrival of the new job $s_*^{\text{NEW}}$ to the end of the execution of $\pi$.

### Continual Planning for Online Job Arrival

In this work we consider COP tasks with particular properties making the achievement of planning goals arriving online akin to the achieval of "jobs" as in scheduling problems. Namely, (i) executing plans for previous goals should not preclude the possibility to achieve new goals; and (ii) achieving new goals should not necessitate deleting previous ones.

**Definition 3.** A **continual planning for online job arrival (COJA) task** is a COP task $(V, A, c, s_*^{\text{OLD}}, s_*^{\text{NEW}}, s_0, \pi_{s_0, s_*^{\text{OLD}}})$ with the following two properties.

(i) **recoverable states**: for every state $s'$ reached from a state $s$ with an action sequence $\vec{\alpha}$, there exists an action sequence $\overleftarrow{\alpha}$ so that $s'[\![\overleftarrow{\alpha}]\!]$ agrees with $s$ on all variable values that appear as preconditions or goals in the task.

(ii) **stable goals**: for every state $s$ from which $s_*^{\text{NEW}}$ can be achieved, there exists a minimum-cost action sequence $\alpha$ doing so without ever changing the assignment $s \cap s_*^{\text{OLD}}$.

Restriction (i) relates to known notions of invertibility and undoability (e.g. (Hoffmann 2005; Daum et al. 2016)). It serves two purposes. First, it allows to err in the prediction of when re-planning will terminate (and thus what the new initial state will be). If the re-planning takes longer than estimated, then the plan execution will have arrived at a state $s_j$ behind the new initial state $s_i$ used by the new plan. Recoverable states allow to nevertheless use the new plan, by going back from $s_j$ to (a state subsuming) $s_i$ first. Second, (i) alleviates necessary, or accidental, conflicts between the previous goal $s_*^{\text{OLD}}$ vs. the new goal $s_*^{\text{NEW}}$. It may, in general, happen that the new plan temporarily deletes $s_*^{\text{OLD}}$ (e.g. in the Blocksworld if $s_*^{\text{NEW}}$ requires to move a block at the bottom of a stack). Given (i), re-achieving $s_*^{\text{OLD}}$ is always possible.

Stable goals (ii) demand that at least one optimal plan for $s_*^{\text{NEW}}$ does not delete whichever parts of $s_*^{\text{OLD}}$ are already achieved. This restriction is sensible as it excludes *necessary* conflicts between the previous vs. the new goals, i.e., situations where achieving $s_*^{\text{NEW}}$ necessarily involves deleting $s_*^{\text{OLD}}$. The optimality requirement makes sure that deleting $s_*^{\text{OLD}}$ can be avoided without a cost penalty.

Even with (ii), it may of course happen that parts of the previous plan, executed during re-planning in our approach, have to be un-done later on. Furthermore, the re-planning process may not find a minimum-cost plan, or for other reasons return a plan deleting $s_*^{\text{OLD}}$. Such *accidental* goal conflicts are, however, qualitatively different from the necessary ones excluded by (ii). That said, stable goals are not a strict requirement of our approach, but merely a "nice to have" property. Indeed, one of our benchmark domains does not satisfy (ii).

Many applications have recoverable states and stable goals. Examples include warehouse logistics, abstract encodings of Mars rover control, and various types of manufacturing problems. Hoffmann (2005) specifies syntactic criteria allowing to identify tasks with recoverable states, and, given an action sequence $\vec{\alpha}$, to quickly find the recovery sequence $\overleftarrow{\alpha}$.

In our experiments, we focus on domains where each action has an immediate inverse action, and thus the cost of $\overleftarrow{\alpha}$ equals that of $\vec{\alpha}$. This simplifies matters as, given $\vec{\alpha}$, the cost of the recovery sequence is known exactly. It remains a topic for future work to drop this assumption (e.g. drawing on Hoffmann's criteria as just mentioned).

---

**Algorithm 1** SRE

1: **procedure** SRE($s_0, s_*^{\text{OLD}}, h, \pi_{s_0, s_*^{\text{OLD}}}, s_*^{\text{NEW}}, R$)
2:     $\gamma \leftarrow 0$
3:     open $\leftarrow \{(r, r) \mid r \in R\}$
4:     closed $\leftarrow \emptyset$
5:     **while** open $\neq \emptyset$ **do**
6:         $\gamma \leftarrow \gamma + 1$
7:         $(s, ref_s) \leftarrow \arg\min_{(m, ref_m) \in \text{open}} f(m, ref_m, \gamma)$
8:         **if** $(s_*^{\text{OLD}} \cup s_*^{\text{NEW}}) \subseteq s$ **then**
9:             **return** *path to s*
10:        closed $\leftarrow$ closed $\cup \{(s, ref_s)\}$
11:        **for** $m \in$ successors($s$) **do**
12:            $ref_m \leftarrow ref_s$
13:            **if** $((m, ref_m) \notin (\text{open} \cup \text{closed})$ or $g(ref_m, m) < g_{old}(ref_m, m))$ **then**
14:                open = open $\cup \{(m, ref_m)\}$
15:    **return** *fail*

16: $f :: (m, ref_m, \gamma) \mapsto g(s_0, ref_m) + g(ref_m, m) + h(m) + overshot(m, ref_m, \gamma)$

## Simultaneous Re-Planning and Execution

We now describe our simultaneous re-planning and execution algorithm SRE, which is an extension of A* to solve COJA tasks. We first specify the algorithm (and how it relates to A*), then we discuss its theoretical properties.

### Algorithm

Algorithm 1 shows the pseudocode of the SRE algorithm. Its structure closely resembles the structure of A*, and the important differences in the pseudocode are highlighted in red.

In contrast to A*, SRE uses a set of potential starting nodes, which we call *reference nodes* and which are given to the algorithm as a parameter $R$. These starting nodes are different "guesses" on which state the agent will be in when the planning finishes. Each reference node is a potential last state of the current plan towards $s_*^{\text{OLD}}$ before deviating from it (the current plan is also given as a parameter $\pi_{s_0, s_*^{\text{OLD}}}$).

The open list (*open*) is initialized using the reference nodes (line 3). Each element of *open* is a pair containing the search node and its corresponding reference node (the node at which it deviates from the original plan). Each newly created search node retains the reference node of its parent (line 12).

Like in A*, nodes in the open list are expanded in a best-first order according to a scoring function $f$, and put into the closed list afterwards. When a node is expanded, its successors are inserted into the open list if they are new or are reached with a lower $g$-value than before (line 13).

Line 8 shows the termination condition. Following Definition 2, we must check whether both the original goal $s_*^{\text{OLD}}$ and the new goal $s_*^{\text{NEW}}$ are achieved.

Finally, the most important difference is the ordering function $f$ for the open list (line 16). The modified $f$-

function has three parameters[1]: a node, its reference node, and the number of expansions made by the algorithm so far, $\gamma$. The $f$-function assigns a score to a pair $(m, ref_m)$ based on three parts. The first part is $g(s_0, ref_m)$.[2] It represents the time required to move from the initial node $s_0$ to the reference node that was used to reach $m$. The second part is $g(ref_m, m) + h(m)$, the same as A*'s $f$-function. It represents the time needed to get from the reference node $ref_m$ to $m$, the node under consideration, combined with the estimate of the time needed to reach the goal from $m$. The third part depends on the function denoted by $overshot(m, ref_m, \gamma)$. This represents the penalty of having to go back to the correct reference node if the agent has already moved past it before planning finishes. This is possible because in COJA tasks, states are recoverable.

We denote the heuristic function estimating the remaining number of expansions until planning finishes by $\eta(m)$. In order to connect this to the execution time, the number of expansions is multiplied by the time per expansion $t_{exp}$ (Burns, Ruml, and Do 2013).

We define the overshot function with respect to a node $m$, its reference node $ref_m$, and the number of expansions so far $\gamma$. Let $\vec{\alpha}$ be the subsequence of actions on $\pi_{s_0, s_*^{\text{OLD}}}$ taking the agent from the reference node $ref_m$ to the state in which it would be at time $(\gamma + \eta(m)) \cdot t_{exp}$ if planning is estimated to end after the execution reaches $ref_m$, and an empty sequence otherwise. Let $\overleftarrow{\alpha}$ be the recovery sequence of $\vec{\alpha}$. The overshot is then defined as $overshot(m, ref_m, \gamma) = C(\vec{\alpha}) + C(\overleftarrow{\alpha}) + \max((\gamma + \eta(m)) \cdot t_{exp} - C(\pi_{s_0, s_*^{\text{OLD}}}), 0)$.

The overshot is $0$ if the planning is estimated to finish before reaching the reference node $ref_m$. Otherwise, it describes the additional execution time incurred by moving past the reference node and back. If planning takes longer than total execution of $\pi_{s_0, s_*^{\text{OLD}}}$, then the agent will additionally have to wait in $s_n$, the last node of $\pi_{s_0, s_*^{\text{OLD}}}$ (this is described by the last term of the overshot function).

Consider the following illustration:

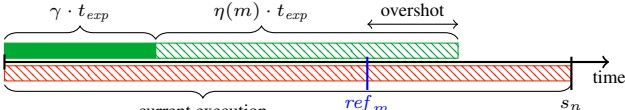

The red dashed bar denotes the time needed to execute the current plan leading to $s_n \subseteq s_*^{\text{OLD}}$. The green bar labeled by $\gamma \cdot t_{\text{exp}}$ is time spent planning so far and the dashed green bar ($\eta(m) \cdot t_{\text{exp}}$) shows the estimation on when the planning will finish. In the illustration, the planning time is estimated to exceed the time when the selected reference node $ref_m$ is reached. The overshot describes this additional execution time, plus the time it takes to go back to $ref_m$.

Having $\gamma$ as an argument for $f$ has an interesting consequence: it now matters when the function $f$ is evaluated for the relative order of the nodes in *open*. In practice, we do not re-evaluate $f$ on all the nodes in the open list each time the

---

[1]$s_0$ is treated as a default parameter

[2]In SRE the $g$-function takes two arguments, and returns the cost (time in our context) from the first to the second. In A* this is implicit as it is only used to denote the cost from the initial node.

best element is retrieved (line 7). Instead, we approximate the value of $f$-function by keeping the search nodes sorted only by $g + h$, but separately for each reference node. Subsequently, we do the full evaluation only to select the next reference node for which a node should be expanded using the nodes with minimal $g + h$ for each reference node. This approximation is justified by the fact that a changed value of $\gamma$ affects all the nodes corresponding to the same reference node equally. The loss of precision comes from disregarding differences in $\eta$.

Coming back to the classical A* formulation, note that there is a parallelism between $g$ and $\gamma \cdot t_{\text{exp}}$ (execution time and planning time so far) as well as between $h$ and $\eta \cdot t_{\text{exp}}$ (estimated time till the end of execution and planning, respectively). There is an important difference though: while exploring a node will not influence the $g$ value of other nodes, $\gamma$ will change its value for all nodes expanded in the future. Note additionally that the *true value* of function $g$ (usually denoted by $g^*$) does not depend in any way on heuristic $g$. In contrast, how many expansions are needed until the end of planning (denoted by $\eta^*$) depends on the heuristic function $\eta$.

## Theoretical Analysis

For A*, it can be shown that the algorithm finds an optimal solution, provided that the heuristic function is admissible (and nodes can be reopened). A similar guarantee can not be given for SRE. The essential difference between the two settings (and thus necessarily between the two algorithms) is that for a classical planning task, the exploration of the state space during the planning phase comes at no cost. On the other hand, in an online setting, exploring a part of the search space that is not going to be used in the solution can decrease the quality of the final plan, since that time was not used effectively. Therefore, unless the heuristic functions $\eta$ and $h$ were perfect, there is no guarantee that SRE will find an optimal solution. We are, however, able to prove that SRE's stopping policy is the correct one. Moreover, in this section, we revisit the baselines mentioned in the introduction and analyze the circumstances under which they can outperform SRE.

SRE stops the search as soon as the first state compliant with both of its goals is found, which raises the question if there is some trade-off between continuing the search and the quality of the solution. We show that continuing the search can not result in a better plan, assuming the heuristic functions $h$ and $\eta$ are admissible.

We will use $h^*(m)$ to denote the true value of the cost to reach the goal from $m$, and $\eta^*(m)$ to denote the number of expansions from node $m$ to the end of planning. Following the same notation style, $f^*(m, ref_m, \gamma_m)$, and $overshot^*(m, ref_m, \gamma_m)$ denote functions $f$ and $overshot$ calculated using $h^*(m)$ and $\eta^*(m)$ instead of the heuristics $h$ and $\eta$. We are using the notation $\gamma = \gamma_m$ to indicate that the third argument of the $f$-function is the value of $\gamma$ when the node $m$ was explored.

**Theorem 1.** *Let $h$ be admissible with respect to planned execution time and $\eta$ admissible with respect to number*

*of expansions. Additionally, assume that for the path $\alpha$ that is a prefix of the path $\alpha'$ it holds $C(\vec{\alpha}) + C(\overleftarrow{\alpha}) \leq C(\vec{\alpha'}) + C(\overleftarrow{\alpha'})$ (well-behaved recovery paths). Let $\sigma_1 = \overline{s_0 s_1 \ldots s_i p_1 p_2 \ldots p_m}$ be the sequence of states corresponding to the first solution $\pi_1$ found by SRE ($s_i$ is the reference node and $p_m$ is the final state of $\sigma_1$). Assume the algorithm continued the search and found another solution, with its sequence of states being $\sigma_2 = \overline{s_0 s_1 \ldots s_j q_1 q_2 \ldots q_n}$ ($s_j$ is the reference node and $q_n$ is the final state of $\sigma_2$). It holds that $f^*(p_m, s_i, \gamma_{p_m}) \leq f^*(q_n, s_j, \gamma_{q_n})$*

*Proof.*

$$f^*(p_m, s_i, \gamma_{p_m}) =$$
$$= g(s_0, s_i) + g(s_i, p_m) + overshot^*(p_m, s_i, \gamma_{p_m})$$
$$= f(p_m, s_i, \gamma_{p_m}) \tag{1}$$
$$\leq f(q_l, s_j, \gamma_{p_m}) \tag{2}$$
$$= g(s_0, s_j) + g(s_j, q_l) + h(q_l) + overshot(q_l, s_j, \gamma_{p_m})$$
$$\leq g(s_0, s_j) + g(s_j, q_l) + h^*(q_l) + overshot^*(q_l, s_j, \gamma_{p_m}) \tag{3}$$
$$\leq g(s_0, s_j) + g(s_j, q_l) + h^*(q_l) + overshot^*(q_l, s_j, \gamma_{q_l}) \tag{4}$$
$$\leq g(s_0, s_j) + g(s_j, q_n) + overshot^*(q_n, s_j, \gamma_{q_n}) \tag{5}$$
$$= f^*(q_n, s_j, \gamma_{q_n})$$

The true cost of the solution $\pi_1$ is $f^*(p_m, s_i, \gamma_{p_m}) = g(s_0, s_i) + g(s_i, p_m) + overshot^*(p_m, s_i, \gamma_{p_m})$. Following the search structure of SRE, at some point we chose to expand $p_m$. Since $p_m$ is the last node on the path and our heuristic functions are admissible, the true cost $f^*$ is equal to the cost function $f$ (equality 1). Inequality 2 comes from our choice of the node $p_m$ over some node $q_l$ from $\sigma_2$. The admissibility of function $\eta$ and the assumption of well-behaved recovery paths yields the admissibility of the function $overshot$. Having $h$ and $overshot$ admissible with respect to $h^*$ and $overshot^*$, we get inequality 3.

If the overshot would be calculated at some later point $\gamma_{q_l}$ when exploring $q_l$, its value would be greater or equal to the value at time point $\gamma_{p_m}$ (inequality 4). Finally, inequality 5 results from $\gamma_{q_n} + \eta^*(q_n) = \gamma_{q_l} + \eta^*(q_l)$ and the fact that $g(s_j, q_l) + h^*(q_l) \leq g(s_j, q_n)$. □

We initially described SRE as A$^*$ with multiple potential starting nodes that accounts for planning times. A different intuition can be obtained by thinking of SRE as similar to running multiple instances of A$^*$ with different initial states, and giving them computation time depending on the value of their $f$-function, combined with reasoning about the time that has passed and the time needed to finish the search. However, different search instances may influence each other, as the time passes for all of them simultaneously and thus affects their reasoning about planning time.

In SRE, the set of reference nodes $R$ is considered to be supplied by the user (a parameter the user can adjust depending on the application). If $R$ would have many elements (e.g., all the nodes of the original plan), it would saturate the processor, but the decision on when to deviate from the original plan would not be limited by sparsity of the set of reference nodes. If $R = \{s_n\}$, containing only the last node of the original plan, then SRE collapses to the baseline that always finishes execution before starting a new plan, thus missing out on the opportunities to deviate from the original plan to make progress towards the new job.

Planning for only one node limits the possibilities the agent has, but focuses the effort (there is no split attention between paths from different reference nodes). Thus, even though SRE offers the advantage of deviating from the original path sooner and finding a better path that way, there is no guarantee it will always outperform the baseline. Consider a scenario in which $R = \{r, s_n\}$ and the optimal path starts from $s_n$. Furthermore, the planning time, if planned only for $s_n$ as the initial node, is exactly the time that the agent will take to execute the original path (so the baseline does not have any waiting time in $s_n$). If at any point, due to imprecise heuristic functions, SRE would explore a node with $r$ as its reference node, the time would be irretrievably lost and the agent would have to wait in $s_n$ until planning is finished.

The same is true for the second baseline we are considering: stopping the execution immediately when a new job arrives. If moving any further along the original plan is getting the agent further away from the new goal (and the old goal may also be achieved on a plan towards the new one), SRE will be outperformed as it will have to move back eventually.

Having noted the situations in which the baselines outperform SRE, outside the edge cases, SRE's parallel planning and execution on the one side, and the flexibility in choosing when to deviate from the original plan on the other side, makes it better suited for tasks with jobs arriving online.

## Experiments

We implemented SRE in Fast Downward (Helmert 2006). In our implementation, we use a standard A$^*$ open list for each reference node, using the SRE extensions to the $f$-function only to select the open list to be used for the next expansion to avoid having to re-sort the open list. When overshooting a reference node, our implementation assumes that each action has an inverse action with the same cost.

Like Bugsy (Burns, Ruml, and Do 2013), we estimate the remaining number of expansions as $\eta(m) = delay * d(m)$ (Dionne, Thayer, and Ruml 2011), where *delay* is the (moving) average number of expansions between inserting a node into the open list and expanding it, and $d$ is an estimation of the remaining steps to the goal (like $h$, but ignoring action costs) under node $m$. The expansion delay is important to counteract *search vacillation* (Dionne, Thayer, and Ruml 2011), referring to the search fluctuating between different solution paths and, in our case, potentially of different reference nodes.

Our key performance metric is the *total time*, i.e. overall time for planning and execution. We are using an instance-specific factor to translate plan cost into execution time as a number of expansions, so the total time is also measured in number of expansions.

In all experiments, the popular FF heuristic (Hoffmann and Nebel 2001) is used to guide the search. For the expan-

sion delay, we use a moving average over the last 100 expansions. The experiments were run on a cluster of Intel Xeon E5-2660 CPUs with a clock rate of 2.20 GHz. The time and memory limits were set to 30 minutes respectively 4 GB.

### Benchmarks

We adapted the IPC domains Elevators, Logistics, Rovers, Tidybot, Transport, and VisitAll to our setting, as representatives of applications where goals have a job-like nature in the sense of (i) recoverable states and (ii) stable goals. We included some variance though to test borderline situations. Elevators, Logistics, Transport, and VisitAll satisfy (i) and (ii), plus the additional assumption that an action sequence $\vec{\alpha}$ and its recovery sequence $\bar{\alpha}$ have the same cost. Rovers also satisfies (i) and (ii), but not the same-cost assumption: actions like taking an image don't need to be inverted. In assuming the opposite, our implementation is pessimistic which may adversely affect the plan cost reported. In Tidybot, finally, there are cases where objects are placed behind each other, and the robot cannot reach behind the object in the front. We added an "un-finish" action to ensure (i) recoverability. However, previously finished objects must be picked up again in these cases. Thus the nice-to-have condition (ii) is not satisfied.

The instances were adapted by splitting the set of goals in two: the first half is available in the beginning, and the other one becomes available later. The second set of goals is scheduled to appear during the execution of the first computed plan to obtain interesting instances. Since we are interested in a combination of planning and execution time, we need to convert both into the same unit.

A run of SRE on one such instance will look as follows:

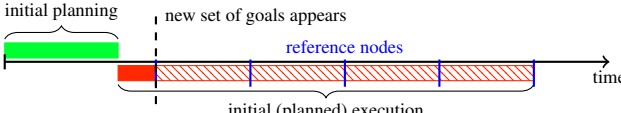

The initially computed plan is being executed as a new job arrives. Here, the planner considers 5 reference nodes as potential initial states for the new plan.

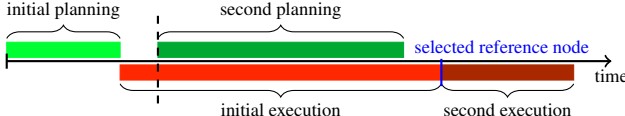

The planner has computed an updated plan that starts from the second to last reference node. The initially computed plan is executed until that point before switching to the new plan. The total time is the time from the start of the first planning phase to the end of the overall execution.

In order to obtain interesting benchmark instances, we tried to ensure that the second planning phase starts and ends during the first planned execution. Thus, we let the second set of goals appear after a fraction of $0.1$ of the initial plan is executed. We estimated the length of the second planning phase by running the planner offline with all goals enabled,

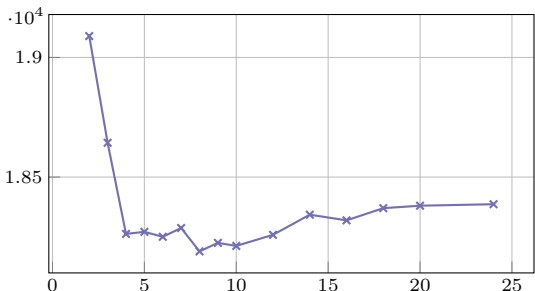

Figure 1: Total time as geometric mean over all instances (Y-axis) for SRE with different numbers of reference nodes (X-axis).

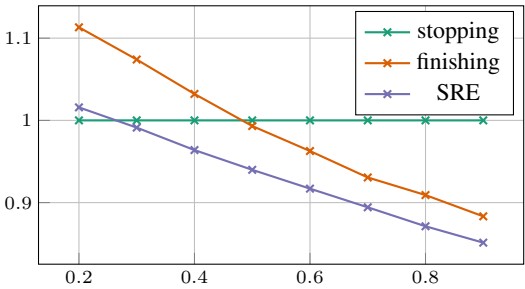

Figure 2: Total time as geometric mean over all instances relative to stopping and re-planning immediately (Y-axis) for $E = 0.2, 0.3, \ldots, 0.9$ (X-axis).

and used that to generate instances where the second planning phase is estimated to end at $E = 0.2, 0.3, \ldots, 0.9$ of the initially planned execution. This is achieved by adjusting the factor for the translation of the action cost to execution time, thereby changing the duration of the initially planned execution.

### Results

SRE has one important parameter: the selection of the reference nodes. In our implementation, we set a number of reference nodes $n_R$, which are then selected in uniform intervals from the current plan. Figure 1 shows the total time (in number of expansions) for different values of $n_R$ across our full benchmark set. If there are too few reference nodes, the algorithm does not have the best starting point for the next plan available. On the other hand, the performance also decreases slightly if too many reference nodes are used, as it becomes more difficult to settle on the most promising one quickly (especially if the planning time estimation is not very accurate). On average, SRE chooses nodes for expansion corresponding to the reference node which is used for the solution $34\%$ of the time, more for fewer reference nodes ($43\%$ for $n_R = 3$), and less the more reference nodes are used ($28\%$ for $n_r = 24$). The overall best results are obtained with $n_R = 8$, and we use that setting for the remaining experiments.

We compare SRE to the two baselines: (a) finishing execution while planning only for the new goals and (b) stop-

ping execution and re-planning immediately. Figure 2 shows the results for different expected end points of the second planning phase, as total time relative to the performance of stopping and re-planning immediately. If the planning time is very short compared to the execution time (small values of $E$), stopping works well. However, if planning is non-trivial ($E \geq 0.3$), SRE performs better. Furthermore, SRE always outperforms baseline (a), for all values of $E$ on all domains. On average, SRE reduces the total time by $6.9\%$ compared to stopping and re-planning immediately, and by $5.6\%$ compared to finishing the planned execution. The results are similar across all domains, except that the relative strength of the baselines differs. On Transport and VisitAll, stopping is better than finishing for $E \leq 0.6$ respectively $E \leq 0.7$, though SRE is the best algorithm for $E \geq 0.4$. On Rovers, stopping is only better than finishing for $E = 0.2$, and SRE is the best algorithm for all values of $E$. The biggest advantage over both baselines is obtained in Elevators, with a total time reduction $7.1\%$ and $7\%$ over stopping and finishing respectively.

Both baselines waste time, though in different ways. Halting the execution is inefficient as the agent is idle while planning. Finishing the execution exploits the parallelism of proceeding with the execution. However, planning only for the second set of goals is usually quite fast, and there would be more time available while waiting for the execution of the initial plan to finish. SRE uses this time more efficiently to compute better overall plans, and effectively improves the combined planning and execution time over both baselines.

## Conclusion

Many planning applications feature the arrival of new jobs while a plan is already being executed. We introduced an algorithm, SRE, which solves this problem effectively: planning simultaneously for multiple potential initial states while proceeding with the execution. The algorithm is aware of its own planning time to select such an initial state in an informed manner. On a set of planning benchmarks, SRE clearly outperforms both vanilla solutions, (a) finishing execution prior to executing the new plan, and (b) stopping execution and waiting for re-planning to terminate.

An interesting question for future research is whether our approach can be extended to, and be useful in, situations with unrecoverable states, i.e., where goals may be in direct conflict. We also believe that our ideas may be brought to bear on domain-specific solutions to achieve better performance, for example in warehouse logistics.

## Acknowledgements

This research was sponsored in part by the ERC Synergy project 610150 (ImPACT) and by the German Research Foundation (DFG) under grants HO 2169/5-1, "Critically Constrained Planning via Partial Delete Relaxation", and 389792660 as part of TRR 248 (see https://perspicuous-computing.science).

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
