# OpenReview forum: "Simultaneous Re-Planning and Plan Execution for Online Job Arrival"
_icaps-conference.org/ICAPS/2019/Workshop/HSDIP_

### Official Review · AnonReviewer2 · 2019-03-28
**Very well written, solid paper**

**Rating:** 9
**Confidence:** 4

**Review:**

The paper introduces a search algorithm for situations where new goals
appear during the execution of a plan and the time to plan has to be
weighed against the time to execute the plan. The algorithm (SRE) plans
while executing the plan for the previous goals. It looks for a plan
that achieves the current and the next goal simultaneously starting
from one of a set of reference states that will be reached during the
execution. The time to reach these reference states (or to go back to
them if they were already passed) is taken into account when selecting
which node to expand next in the search.

The paper is extremely well written and a pleasure to read. The problem
itself is interesting and a good fit for HSDIP.

On the theoretical side, the authors argue convincingly that the online
nature of the problem makes it impossible to give a guarantee as strong
as the one for A* but they do prove that the first solution discovered
by SRE provides a better trade-off than solutions that would be found
when SRE would keep searching, so it is correct to stop SRE with the
first discovered solution. While I do have some questions about details
of this proof (see below), I am reasonably convinced that it is correct
in the setting where it is used.

In the experimental section, the paper demonstrates a clear benefit of
SRE over the baseline approaches of stopping the current execution to
plan for the next goal or finishing execution while planning for the
new goals. The experiments are generally well designed, although I find
the choice of the (inadmissible) FF heuristic strange after the
theoretical section focused only on admissible heuristics.


Questions and comments about the proof:

1) I think the admissibility of h + overshot is only implied if
C(\alpha->) = C(\alpha<-). Otherwise, underestimating \eta(m) could
give a shorter \alpha-> which has a more expensive \alpha<-.

2) Should any of the g in the proof be g^* instead? If so, what is the
argument that the g values are optimal when SRE expands the node? If
not, why not?

3) Can't \gamma_{q_n} + \eta^*(q_n) be larger than \gamma_{q_l} +
\eta^*(q_l) because there can be expansions between q_l and q_n that
are unrelated to this plan?

4) Does the equality g(s_j, q_l) + h^*(q_l) = g(s_j, q_n) assume that
the path from s_0 to q_n through q_l is optimal?



Minor comments:
* The statements
    "It may, in general, happen that the new plan temporarily deletes
    s^OLD_*"
  and
    "at least one optimal plan for s^NEW_* does not delete whichever
    parts of s^OLD_* are already achieved"
  seem to contradict each other. It would be good to add a clarifying
  statement.

* The illustration of the overshot function confused me at first. For
one, parts of the function are explained before the function is
introduced, so I didn't look for an explanation of the maximization
after the illustration. I suggest moving the sentence after the
illustration before the function definition. In addition, the
illustration only contains the execution of the current plan, so I
cannot find \alpha-> and \alpha<- in there. I suggest to add a third
bar to the plot that shows the potential execution of the plan through
m with marks for s_0, ref_m, ref_m[\alpha->], ref_m[\alpha->,\alpha<-],
m, and the final state of the new plan. it might also help to say
(before the illustration) that this only illustrates the case where the
planning ends after reaching ref_m but before reaching the end of the
execution. You could also add illustrations for the other cases (there
is enough space) but I don't think it is important.

* There is an additional "=" in the first line of the proof.

* I believe the reference for Fast Downward should be from JAIR, 2006.

---

> ### Author Response · Authors · 2019-04-10
> **thanks for the review**
>
> Thank you for the suggestions and nice catches in the proof. We have submitted an updated version of the paper with fixes to the proof, and will improve unclear parts for the final version.
>
> We consider an admissible heuristic in Theorem 1, but we give up on those strict requirements in practice. In our experiments, we already lose these guarantees with the inadmissibility of the estimation for the remaining expansions, so we don't insist on an admissible heuristic either.
>
> Regarding the questions and comments on the proof of Theorem 1:
>
> 1) You are right - the admissibility of h+overshot does not hold for every recovery path. (Thanks for noticing!)
> An additional assumption is needed. It might be either
>   a) (as you suggested) C(\alpha->) = C(\alpha<-),
>   b) or if \alpha-> is a prefix of \alpha'-> it should imply that C(\alpha->) + C(\alpha<-) <= C(\alpha'->) + C(\alpha'<-)
> We think that option b) is a reasonable assumption, and have revised the proof accordingly.
>
> 2) We believe that g should not be replaced by g* in any place. We are assuming that by g*(x,y) you mean "optimal path from x to y". In the steps of the theorem we never discuss the optimality of the path found so far, but compare two different paths that the algorithm has found.
> In the steps of the proof:
>   - the second line is the definition of f* (the same as f, but using \eta* and h* instead of \eta and h. The heuristic is 0 as this is the last node). In that line, g(s_0, s_i) is the cost of the part of the original plan from s_0 to s_i, and g(s_i, p_m) is the cost of the path found by the algorithm from s_i to p_m. That path might or might not be optimal in terms of cost.
>   - the equality after (2) is by the definition of f, with the first two terms denoting paths found so far.
>   - the equality (5) separates g(s_j, q_n) into two parts (that were found by the algorithm).
>
> 3) The \eta function estimates the overall number of expansions, including those unrelated to the plan.
> Its real value, \eta*, takes them into account as well.
>
> 4) You are right, an additional assumption is needed for the equality to hold. We should assume that the optimal path from q_l to q_n would be found by our search algorithm and that q_n is the "closest" node compliant with the goal (only then would h*(q_l) be equal to g(q_l, q_n)). Instead of introducing those additional assumptions, we propose replacing the equality with inequality (i.e. h*(q_l) is smaller or equal to g(q_l, q_n)).

---

> > ### Comment · AnonReviewer2 · 2019-04-11
> > **Thanks for the clarifications**
> >
> > 1) I agree that b) makes for a better assumption and fixes the issue. Thanks for changing it in the paper immediately. In the paper you were slightly stricter than your comment and used a strict inequality which prevents 0-cost paths. Is that necessary?
> >
> > 2) Thanks for the clarification, I misread f^* to mean the optimal path cost in one step.
> >
> > 3) I see, thanks for clarifying this as well.
> >
> > 4) I agree that replacing the equality with an inequality makes sense here.
> >
> > That clears up all my questions. I'm still strongly in favor of accepting the paper.

---

> > > ### Author Response · Authors · 2019-04-11
> > > **It should not be strict inequality**
> > >
> > > The strict inequality was a lapsus, we'll fix that for the final version. Thanks!

---

### Official Review · AnonReviewer1 · 2019-04-02
**Well-written paper tackling the interesting problem of re-planning during plan execution.**

**Rating:** 8
**Confidence:** 4

**Review:**

This paper presents a new algorithm, called simultaneous re-planning and
execution (SRE) designed for planning problems where additional goals/jobs
arise during the plan execution of a plan for the original problem. In this
continual online planning (COP) setting, SRE allows a trade-off between the
two naive solutions that would be to stop plan execution for re-planning or to
finish plan execution entirely before re-planning. The assumption of the COP
setting is that additional jobs do not conflict with the original goals and
that states are recoverable (some form of invertibility).

This paper addresses an interesting topic relevant for the workshop and
provides both an algorithmical and an empirical contribution for the COP
scenario. It is also well written and mostly easy to follow, so a clear accept
for the workshop.

I had some initial difficulties understanding Definition 2 in the described
context of online re-planning and plan execution: the task definition assumes
the existence of a plan for the old goal already (that must be computed as part
of the entire scenario as I understood it), so that it seems to me it does not
capture the entire setting, but rather the setting from the moment on where the
additional goals arrive. I also initially expected that there would be an
entire series of new goals arriving, and thus a series of re-planning steps
occurring. I think that the exact setting should be described more concisely
already early in the paper. For example, I found the explanations in Section
"Experiments" in the first column of page 6 very helpful to understand not only
the assumptions of the COP setting made in this paper, but also the precise
problem being tackled.

Probably for a similar reason, I found the first illustration in column 1 of
page 4 difficult to understand: what does the green part represent and what is
node s_n? I understand that overshot is the time due to reaching the reference
node ref_m earlier than estimated, but I don't see how that figure illustrates
the "the additional execution time incurred by moving past the reference node
and back", and it also doesn't help to understand the definition of the
overshot itself. What does the "- C(\pi_s_0, s_*^OLD)" account for? Is that the
"last term of the overshot function" that describes the waiting time of the
agent in s_n?

Finally, I suggest to include more heuristics both for guiding search and
estimating the remaining number of expansions in the experiments. Generally, I
understand that one needs to fix some choices when introducing and evaluating a
new setting and a new algorithm, but I would expect that including more
heuristics would not cause too much extra work and increase the weight of the
results and their interpretation. (In contrast to that, leaving some of the
assumptions on the types of problems behind would presumably be much more
challenging.)

Some minor comments:
- Do not give copyright to AAAI press. Use \nocopyright in the preamble.

- Helmert 2011: this reference is wrong (I really hope that there is no CoRR
abs copy of the original JAIR 2006 paper that should be cited here)

- Def 2: why call "s^old" the "current" goal and not the old one (or rename
s^old to s^curr)?

- Algorithm 1 (and other places): is there a reason/intuition to use letter "m"
for nodes?

- \eta: I assume this function estimates the remaining number of expansions for
*planning* and not for *execution* of the previous plan until planning
finishes, right?

---

> ### Author Response · Authors · 2019-04-10
> **thanks for the review**
>
> Thank you for the detailed comments, especially for pointing out the places that caused difficulties in understanding. We will clarify the problem setting and overshot explanations in the final submission.
>
> We agree that results for more heuristics are interesting and consider doing a more extensive study in future work. However, generating the instances for that heuristic is not straightforward. The instances for the experiments are generated from observations of offline search, so using a different heuristic also requires generating new instances for that heuristic (otherwise, the online goals might be scheduled e.g. after the initial execution already finished, or the second planning phase might take longer than the initial execution, so our approach would not be necessary and collapse to a baseline).
>
> Regarding your last question (under "Minor comments"): yes, \eta estimates the remaining number of expansions until the end of planning. However, we were not able to find the misleading statement in the paper, can you give us the exact location?

---

> > ### Comment · AnonReviewer1 · 2019-04-11
> > **thanks for the reply**
> >
> > Thanks for your reply and promising to expand the things that were unclear to
> > me. If you have any chance to already attempt at explaining the example figure
> > illustrating the overshot and also the mathematical term (my comments in
> > paragraph 4), I would be very happy, but of course this is not necessary at
> > all.
> >
> > However, regarding your other comment, I don't quite understand why you would
> > need to generate instances for a specific heuristic. Probably it would be a
> > good idea to go into more details of how you "adapted" the domains for the
> > current experiments. Can't you just add some of the goals of the original
> > problem as later arriving goals, for example? And why is the proces not
> > heuristic-agnostic?
> >
> > I cannot find the reason why I found unclear what \eta(m) means, so please just
> > ignore this comment.

---

> > > ### Author Response · Authors · 2019-04-12
> > > **overshot and benchmark generation clarifications**
> > >
> > > The overshot accounts for the additional time needed to go back to the reference node (for which the plan is computed) from where the agent is estimated to be when planning finishes. Consider the following cases:
> > >
> > > a) the estimated position is before the reference node (on the original path):
> > >   In this case, the total overshot is 0. \alpha-> and \alpha<- are by definition empty sequences. The third term is 0 as well because the estimated position is before the reference node, and thus also before the last node on the path.
> > >
> > > b) the estimated position will be after the reference node but before the last node (of the original path):
> > >   This is the case shown on the illustration. The dark green color (\gamma * t_exp) denotes the past planning time. Striped green denotes the expected duration for the planning procedure (\eta * t_exp). Since the execution would reach the reference node (denoted by ref_m in the illustration) before planning finishes, the agent has to go back. In this case, the overshot is the cost of going from the reference node to the end of the striped green line (C(\alpha->)) and the time needed to go back to the reference node (C(\alpha<-).
> > >
> > > c) the estimated position will be the last node of the original path (s_n), and the agent will stay idle there while waiting for the planning to finish.
> > >   This case finally makes the last term meaningful: the estimated end of planning ((\gamma + \eta) * t_exp) is greater than the cost of reaching the end of the current plan (C(\pi_s_0, s_*^OLD)), so the difference is the waiting time in s_n.
> > >
> > > The remark that s_n is the last state of the planned execution seems to have gone missing in the last revisions, we will make sure to include it again in the final version.
> > >
> > > Regarding the benchmarks:
> > > The goals are split into two halves, the first half is available immediately and the second half is scheduled to appear at a fixed time later. This time is determined by running the planner offline using only the first set of goals, and from the number of expansions N and plan cost C (which is also translated to a number of expansions as execution time), we generate the adapted instance such that the second set of goals appears after 10% of the initially computed plan is executed, i.e. after N + 0.1 * C expansions. Furthermore, we use the number of expansions to solve the entire problem offline as an estimate for the duration of the second planning phase. This is then used to adjust the translation factor for the plan cost to execution time in number of expansions such that in the generated instances, the second planning phase is estimated to end at a fraction of E = 0.2, ..., 0.9 of the execution (i.e. at N + E * C). In the illustrations in the Benchmarks subsection of the Experiments, the bright green bar corresponds to N, the bright red bar corresponds to C, and the dark green bar ends at N + E * C.

---

> > > > ### Comment · AnonReviewer1 · 2019-04-16
> > > > **thanks for the clarifications**
> > > >
> > > > Thanks a lot for the explanation of the figure that illustrates the overshot (and its mathematical definition), this all makes sense.
> > > >
> > > > I think I also more or less understood how you generate the benchmarks, but I still don't understand why using different heuristics would cause so much extra work. Is the process of creating instances with an offline search prior to running the actual algorithm not automated, i.e., does it require a lot of manual adjustment? Or is the reason that the instances are basically tied to the SRE planner, i.e., the instances are not just plain standalone PDDL files?

---

> > > > > ### Author Response · Authors · 2019-04-16
> > > > > **instances**
> > > > >
> > > > > The instances are standalone PDDL files with extensions for online goals and the factor for the execution time. Most of the process to generate the instances is automated. However, since we decided early on to use FF as the only heuristic in our evaluation, we did not automate the complete process, so some manual work is still required.

---

### Meta-Review · Program_Chairs · 2019-04-25

**Recommendation:** Accept
**Confidence:** 5

**Metareview:**

Dear Authors,
thank you very much for your submission. We are happy to inform you that
we have decided to accept it and we look forward to your talk in the workshop.
Please, go over the feedback in the reviews and correct or update your papers
in time for the camera ready date (May 24).
Best regards
HSDIP organizers